# SegMASt3R: Geometry Grounded Segment Matching

**Rohit Jayanti**[1][*][†]    **Swayam Agrawal**[1][*][‡]    **Vansh Garg**[1][*]    **Siddharth Tourani**[2,3]

**Muhammad Haris Khan**[3]    **Sourav Garg**[4]    **Madhava Krishna**[1]

[1]IIIT Hyderabad    [2]University of Heidelberg    [3]MBZUAI    [4]Independent

 Project Page: `https://segmast3r.github.io/`

## Abstract

Segment matching is an important intermediate task in computer vision that establishes correspondences between semantically or geometrically coherent regions across images. Unlike keypoint matching, which focuses on localized features, segment matching captures structured regions, offering greater robustness to occlusions, lighting variations, and viewpoint changes. In this paper, we leverage the spatial understanding of 3D foundation models to tackle wide-baseline segment matching, a challenging setting involving extreme viewpoint shifts. We propose an architecture that uses the inductive bias of these 3D foundation models to match segments across image pairs with up to $180°$ rotation. Extensive experiments show that our approach outperforms state-of-the-art methods, including the SAM2 video propagator and local feature matching methods, by up to 30% on the AUPRC metric, on ScanNet++ and Replica datasets. We further demonstrate benefits of the proposed model on relevant downstream tasks, including 3D instance mapping and object-relative navigation.

## 1 Introduction

Segment matching establishes correspondences between coherent regions—objects, parts, or semantic segments—across images. It underpins video object tracking [17], scene-graph construction [18, 23], robot navigation [16, 15, 32], and instance-level SLAM [26, 44]. Because it matches extended structures rather than sparse, texture-sensitive points, it is more robust to noise, occlusion, and appearance change. Mapping structured regions instead of isolated pixels also boosts interpretability and allows geometric or semantic priors to be injected, enabling higher-level reasoning about scene content.

Segment matching degrades sharply under *wide-baseline* conditions, where images of the same scene are taken from widely separated viewpoints that introduce severe perspective, scale, and up to $180°$ rotation changes [34]. Such cases arise in long-term video correspondence and robotic navigation, and demand models that reason over global 3D structure and enforce geometric consistency. Current approaches that depend on features from pre-trained encoders like DINOv2 [30] or ViT [10] often mismatch repetitive patterns or fail to link drastically different views of the same object.

In this paper, we propose to leverage the strong spatial inductive bias of a 3D foundation model (namely MASt3R [22]) to solve the problem of wide-baseline segment matching. 3D Foundation Models (3DFMs) are large-scale vision models trained to capture spatial and structural properties of scenes, such as depth, shape, and pose. Unlike appearance-focused models, 3DFMs learn geometry-aware representations using 2D and 3D supervision, enabling strong generalization across tasks

---

\* Equal Contribution. [†] Corresponding Author. [‡] Now at Google DeepMind.

39th Conference on Neural Information Processing Systems (NeurIPS 2025).

like reconstruction and pose estimation. Their inductive bias toward spatial reasoning makes them well-suited for applications requiring geometric consistency such as our problem of wide-baseline segment matching.

We adapt MASt3R for segment matching by appending a lightweight *segment-feature head* that transforms patch-level embeddings into segment-level descriptors. Given an image pair, these descriptors are matched to establish segment correspondences. The head is trained with a contrastive objective patterned after SuperGlue [35]. Experiments show that our approach surpasses strong baselines, including SAM2's video propagator which is trained on far larger datasets and state-of-the-art local feature matching methods. Finally, we demonstrate its practical utility in two downstream tasks: object-relative navigation and instance-level mapping.

**Contributions**  Our key contributions are summarized below:

- We introduce a simple but effective approach for learning segment-aligned features by leveraging strong priors from a 3D foundation model (3DFM) MASt3R. A differentiable segment matching layer is employed to align features across views, while a *segment-feature head* transforms dense pixel-level representations into robust segment-level descriptors.

- To address the under-explored problem of wide-baseline segment matching, we construct a comprehensive benchmark comprising both direct segment association methods and those based on local feature matching. Our method demonstrates significant improvements over all baselines on challenging wide-baseline image pairs.

- We validate the practical utility of our approach by applying it to the downstream applications such as 3D instance mapping and object-relative topological navigation. Our method outperforms competitors by significant margins showcasing the efficacy of our design choices.

## 2  Related Work

**Segment Matching and Segmentation Foundation Models.**  Robust segment-level association has emerged as a crucial intermediate step for high-level vision tasks such as scene graph construction, long-term object tracking, and multi-view instance association. While related sub-problems like video instance segmentation and object tracking have been studied extensively in recent works [46, 45, 41, 11, 31, 29, 24, 6], the broader challenge of matching segments across arbitrary viewpoints, modalities, and time remains comparatively under-explored. Large-scale class-agnostic segmentation models have begun to close this gap. The Segment Anything Model (SAM) [20] and its successor SAM2 [34] deliver high-quality masks and include a built-in propagator for associating masks across video frames. However, this propagation module is optimized for short temporal windows and does not explicitly enforce geometric consistency under wide baselines or substantial appearance changes.

**Learning to Match Segments and Overlap Prediction.**  Some recent methods address segment matching more directly. MASA [25] augments SAM's rich object proposals with synthetic geometric transformations to learn instance correspondences, and DMESA [49] extends these ideas to dense matching with improved efficiency. Despite such progress, these approaches are still limited by 2D supervision. An alternative line of work predicts the degree of visual overlap between images [7, 13, 2]. By estimating shared content, these methods implicitly learn region correspondences, yet they also remain confined to 2D training signals.

**Local Feature Matching.**  Sparse [9, 35, 40, 27, 36] and dense [12, 4] local feature matchers propagate pixel-level correspondences that can, in principle, transfer segment labels between views [15, 16]. Nonetheless, like the previous categories, they are trained exclusively on image data and struggle with extreme viewpoint changes.

Across segment matching, overlap prediction, and feature matching, reliance on purely 2D supervision leaves existing techniques brittle under wide-baseline conditions. Our method addresses this limitation by fine-tuning the 3D foundation model MASt3R [22], whose strong geometric priors enable reliable segment correspondence even when image pairs differ by nearly $180°$ in viewpoint.

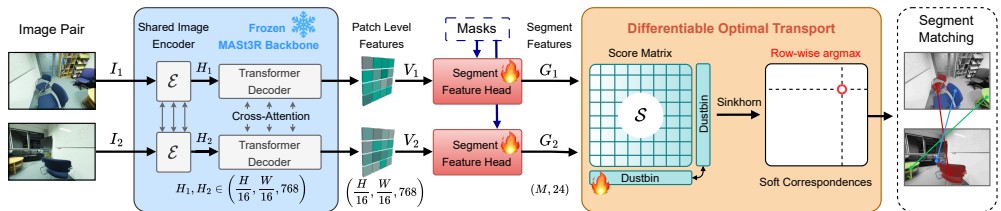

Figure 1: **Pipeline Overview**: An image pair is processed by a frozen MASt3R backbone to extract patch-level features; segmentation masks are obtained either from a parallel segmentation module or ground truth annotation; the patch-level features are aggregated by the segment-feature head to form segment-level descriptors; and these descriptors are then matched across images via a differentiable optimal transport layer to produce segment-level correspondences.

## 3   Method

Figure 1 provides an overview of our method. It builds upon the MASt3R [22] architecture by introducing a Feat2Seg Adapter that maps the patch-level features output by the MASt3R decoder to get segment-level features, which are subsequently matched via a differentiable optimal transport and a row-wise argmax yielding the final segment-level correspondences.

### 3.1   MASt3R Preliminaries

MASt3R is a 3D foundation model pre-trained on a diverse collection of 3D-vision datasets commonly used for tasks such as metric depth estimation and camera-pose prediction. Given a pair of images, it produces dense 3D point maps for each image and identifies correspondences between them. Thanks to training on data with wide-baseline pairs, MASt3R generalizes well to unseen image pairs [22] and consistently outperforms alternative 3D matching methods—such as MicKey [3], which relies on a DINOv2 backbone [3].

**Architecture:**   We summarize here the portions of the architecture we utilize in our pipeline. The two images $I_1$ and $I_2$ are processed in a Siamese manner by a weight-sharing ViT encoder [10] $\mathcal{E}$, resulting in token sets $H_1$, $H_2$, *i.e.*

$$H_1 = \mathcal{E}(I_1), \qquad H_2 = \mathcal{E}(I_2).$$

**Cross-view transformer decoder:**   Next, a pair of CroCo [42, 43] style intertwined transformer decoders jointly refines the two feature sets. By alternating self- and cross-attention, the decoders exchange information between inputs to capture both the relative viewpoints and the global 3-D structure of the scene. We show via an ablation study, these cross-view aware decoders provide a significant boost to the segment matching in Section 5

The resulting geometry-aware representations are denoted $V_1$ and $V_2$:

$$\left(V_1, \ V_2\right) = \mathrm{Decoder}\left(\mathbf{H}_1, \ \mathbf{H}_2\right).$$

Assuming, $H, W$ denote the height and width of the input images, the output geometry-aware patch-level features are of size $(H/16, W/16, 768)$. The original architecture subsequently uses two prediction heads to output dense point-maps, as well as pixel-level features, they are not shown in Figure 1 as they are not used in our pipeline. These portions of the pipeline remain frozen and their outputs are used as is, we instead introduce a new segment-feature head.

### 3.2   Segment-Feature Head: Segment-Aligned Features

The MASt3R decoder produces patch–level embeddings $V_1, V_2$ of shape $(H/16, W/16, 768)$. In the original MASt3R [22], a *feature head* upsamples these tensors to the resolution of the input image. For *segment matching* we introduce another head to transform the patch level features to $M$ segment features. This introduced head is realized as an MLP that upsamples the patch-level

features $(V_1, V_2)$ to image resolution, yielding feature maps of size $(H, W, 24)$, 24 being the feature dimension. We denote this *Feature-to-Segment* head as the **segment-feature** head. In addition to the patch-features, the **segment-feature** takes as input $M$ image resolution segment masks, obtained either from an external segmenter such as SAM2 [34] or from ground-truth annotations. Both the masks and feature maps are flattened along the spatial dimensions yielding flattened tensors.

$$\mathbf{P}_{\text{flat}} \in \mathbb{R}^{24 \times HW}, \qquad \mathbf{M}_{\text{flat}} \in \mathbb{R}^{M \times HW}.$$

To go from pixel-level descriptors to segment descriptors, a single batched matrix multiplication aggregates the pixel descriptors inside each mask:

$$\mathbf{G} = \mathbf{M}_{\text{flat}} \, \mathbf{P}_{\text{flat}}^{\top} \in \mathbb{R}^{M \times 24}. \tag{1}$$

The resulting segment embeddings for the two images are denoted $\mathbf{G}_1$ and $\mathbf{G}_2$ in Figure 1 and are fed to the differentiable matching layer described in Section 3.3. We set $M = 100$ as an upper bound for batch processing. In practice, images typically contain 20-30 masks when training with ground truth annotations; we pad with zeros when fewer masks are present. At inference time, the number of masks can be set arbitrarily, independent of the training-time value of $M$.

## 3.3 Differentiable Segment Matching Layer

The goal of the differentiable segment matching layer is to establish permutation-style correspondences between the $M$ segments from each image, given segment descriptors $(G_1, G_2) \in \mathbb{R}^{(M, 24)}$

**Cosine–similarity affinity.** We first construct an affinity matrix $\mathbf{S} \in \mathbb{R}^{M_1 \times M_2}$ with a simple dot product

$$S_{ij} = \langle \mathbf{g}_i^1, \mathbf{g}_j^2 \rangle, \quad 1 \le i \le M_1, \ 1 \le j \le M_2. \tag{2}$$

$\mathbf{g}_i^1$ and $\mathbf{g}_j^2$ correspond to segment-level features from $G_1$ and $G_2$ respectively. Ideally, segment features corresponding to the same underlying 3D region should have a high similarity score and dis-similar regions a correspondingly low score.

**Learnable dustbin.** Following [35], we incorporate a *dustbin* row and column in the affinity matrix $\mathbf{S}$ to handle segments without correspondences in the other image, which is critical for wide-baseline matching. We augment (2) by concatenating an additional row and column initialized with a learnable logit $\alpha \in \mathbb{R}$, yielding $\tilde{\mathbf{S}} \in \mathbb{R}^{(M_1+1) \times (M_2+1)}$.

$$\tilde{\mathbf{S}} = \begin{bmatrix} \mathbf{S} & \alpha \mathbf{1}_{M_1} \\ \alpha \mathbf{1}_{M_2}^{\top} & \alpha \end{bmatrix},$$

**Soft Correspondences via Sinkhorn** The similarity logits are transformed into a soft assignment matrix $\mathbf{P}$ by $T$ iterations of the Sinkhorn normalisation [38, 37] in log-space:

$$
\begin{aligned}
\mathbf{P}^{(0)} &\leftarrow \exp(\tilde{\mathbf{S}}/\tau), \\
u_i^{(t)} &= \frac{1}{\sum_j P_{ij}^{(t)}}, \quad v_j^{(t)} = \frac{1}{\sum_i P_{ij}^{(t)}}, \\
P_{ij}^{(t+1)} &= u_i^{(t)} \, P_{ij}^{(t)} \, v_j^{(t)}, \quad 0 \le t < T,
\end{aligned}
\tag{3}
$$

where $\tau$ is a temperature hyper-parameter. After convergence, $\mathbf{P} = \mathbf{P}^{(T)}$ is (approximately) doubly stochastic. Throughout all experiments conducted, we assume $T = 50$. The output of the Sinkhorn algorithm $\mathbf{P}^{(T)}$ is a soft bi-stochastic matrix, which has to be discretized to obtain the final segment matches.

**Discrete correspondences.** To obtain the simple final segment matches, we perform a simple row-wise $\arg\max$ over the *non-dustbin* columns:

$$m(i) = \underset{1 \le j \le M_2}{\operatorname{argmax}} P_{ij}, \quad \text{with assignment accepted if } j \ne M + 1.$$

### 3.4 Supervision

**Training objective.** We adopt the SuperGlue cross-entropy loss $\mathcal{L}_{SG}$ [35], extended with explicit terms for unmatched segments:

$$\mathcal{L} = -\sum_{(i,j)\in\mathcal{M}} \log P_{ij} - \sum_{i\in\mathcal{U}_1}\log P_{i,M+1} - \sum_{j\in\mathcal{U}_2}\log P_{M+1,j}, \tag{4}$$

where $\mathcal{M}$ is the set of ground-truth matches and $\mathcal{U}_1, \mathcal{U}_2$ are the unmatched indices in image 1 and 2, respectively. The dustbin parameter $\alpha$ is learned jointly with the rest of the network, enabling the layer to balance match confidence against the cost of declaring non-matches. This fully differentiable design allows the matching layer to be trained end-to-end together with the upstream segment encoders and downstream task losses.

#### 3.4.1 Training Details

The model is trained using AdamW optimizer with an initial learning rate of `1e-4`, weight decay of `1e-4`, and a cosine annealing learning rate schedule without restarts, decaying up to a minimum learning rate of `1e-6` over the full training duration. We use a batch size of 36 and train the model for 20 epochs on a single NVIDIA RTX A6000 GPU. The **segment-feature heads** are initialized with MASt3R's local feature head weights and finetuned further. For the differentiable segment matcher we initialize the single learnable dustbin parameter to $1.0$. The number of Sinkhorn iterations is set to 50. Training our model on ScanNet++ takes 22 hours, whereas a single forward pass during inference with batch size of 1 takes 0.579 seconds.

## 4 Experiments

### 4.1 Datasets

Our network is trained on scenes from ScanNet++ [47] which contain a diverse set of real-world scenes, primarily in indoor settings. We test our model on novel scenes from ScanNet++ as well as perform cross-dataset generalization studies on Replica [39] and MapFree [2] datasets. The former contains high-quality photo-realistic indoor scenes while the later is a challenging outdoor visual-localization dataset, which is sufficiently out-of-distribution considering our training data.

**ScanNet++ [47].** ScanNet++ contains 1 006 indoor scenes captured with DSLR images and RGB-D iPhone streams, all co-registered to high-quality laser scans. The dataset supplies 3D semantic meshes, 2D instance masks, and accurate camera poses, which we utilize in our pipeline. We train on 860 k image pairs from 140 scenes and evaluate on 8 k pairs from 36 validation scenes, sampled with a fixed seed (42) and balanced across scenes and four pose bins: $[0°-45°]$, $[45°-90°]$, $[90°-135°]$, and $[135°-180°]$, defined by the rotational geodesic distance between camera orientations.

**Replica [39].** The Replica dataset contain 18 high-quality, photo-realistic indoor room reconstructions in the form of dense meshes replete with per-primitive semantic class and instance information. In particular, we use a pre-rendered version of this dataset from Semantic-NeRF [50], which directly provides poses, RGB-D sequences, and per-frame semantic masks. For evaluation, we employ the same pose-binning strategy as described above and randomly sample 3200 image pairs across 8 scenes, again ensuring uniform sampling across both scenes and pose-bins.

**MapFree Visual Re-localization [2].** MapFree is a challenging benchmark for metric-relative pose estimation, featuring 655 diverse outdoor scenes (sculptures, fountains, murals) with extreme viewpoint changes, varying visual conditions, and geometric ambiguities. The training split contains 460 scenes with 0.5M images; we uniformly sample 50 scenes yielding 31K image pairs for training. Since test split ground truth is unavailable, we report results on the validation split using 7.8K pairs from 13 uniformly sampled scenes (out of 65 total).

### 4.2 Baselines

**Local Feature Matching (LFM).** Although few models target *segment* matching directly, a wide range of local feature matchers (LFMs) exists at different densities: sparse SuperPoint [9], semi-dense LoFTR [40], and dense RoMa and MASt3R [12, 22]. We harness these state-of-the-art LFMs

| Type | Method | 0°–45° | | | 45°–90° | | | 90°–135° | | | 135°–180° | | |
|---|---|---|---|---|---|---|---|---|---|---|---|---|---|
| | | AUPRC | R@1 | R@5 | AUPRC | R@1 | R@5 | AUPRC | R@1 | R@5 | AUPRC | R@1 | R@5 |
| Local Feature Matching (LFM) | SP-LG [9, 27] | 42.1 | 45.6 | 51.2 | 33.5 | 36.9 | 43.1 | 15.9 | 19.7 | 26.2 | 6.1 | 9.3 | 14.6 |
| | GiM-DKM [36, 11] | 59.1 | 64.9 | 69.7 | 54.9 | 60.2 | 66.1 | 39.6 | 44.5 | 51.8 | 21.3 | 25.9 | 32.7 |
| | RoMA [12] | 61.6 | 68.7 | 73.5 | 58.9 | 66.4 | 73.0 | 47.4 | 56.1 | 65.5 | 30.0 | 39.5 | 49.7 |
| | MASt3R [22] | 59.5 | 68.3 | 74.2 | 57.3 | 65.6 | 72.5 | 52.9 | 60.3 | 68.9 | 45.4 | 52.6 | 62.2 |
| Segment Matching (SegMatch) | SAM2 [34] | 61.9 | 64.6 | 67.5 | 46.6 | 50.1 | 54.0 | 27.9 | 32.5 | 37.2 | 17.0 | 21.6 | 25.4 |
| | DINOv2 [30] | 57.9 | 66.7 | 87.4 | 43.0 | 55.9 | 83.2 | 33.5 | 48.0 | 78.0 | 32.4 | 46.0 | 75.6 |
| | SegVLAD [14] | 44.2 | 58.6 | 81.4 | 32.1 | 49.5 | 76.5 | 23.2 | 42.2 | 70.5 | 20.0 | 39.6 | 66.8 |
| | MASt3R [22] | 51.7 | 54.6 | 69.9 | 45.6 | 49.8 | 68.5 | 41.4 | 47.9 | 69.2 | 39.5 | 48.7 | 72.6 |
| Ours | SEGMASt3R | 92.8 | 93.6 | 98.0 | 91.1 | 92.2 | 97.6 | 88.0 | 89.5 | 96.8 | 83.6 | 85.9 | 95.9 |

Table 1: Performance of selected methods across pose-bins on ScanNet++ [47]. Blue cells mark the best scores; Orange cells mark the second-best.

via the EarthMatch toolkit [5] to obtain segment correspondences. Each matched keypoint pair votes for the source and target masks that contain its coordinates, populating a vote matrix of size $M \times N$ (with $M$ and $N$ segments in the two images). Correspondences are taken as the highest-scoring entries of this matrix; the full algorithm is provided in the supplementary.

**Segment Matching (SegMatch).** We also compute segment matches from dense features of two strong pre-trained vision encoders, DINOv2 [30] and MASt3R [22], as well as SAM2's video propagator [34] to track masks across views. For feature based methods, masks are downsampled via nearest-neighbor interpolation to match feature resolution, and segment descriptors are computed via masked average pooling. Cosine similarity between descriptors yields a match matrix, from which one-to-one links are selected via mutual-check. We further benchmark against SegVLAD [14], which aggregates features from neighbouring segments for segment retrieval based visual place recognition.

### 4.3 Evaluation Metrics

We report two complementary measures of segment correspondence quality - **AUPRC** and **Recall**. *Area Under the Precision–Recall Curve (AUPRC)* integrates precision over the entire recall axis, providing a threshold–independent summary that is particularly informative under the high class–imbalance characteristic of segment matching between image pairs. *Recall@$k$ ($R@k$)* denotes the fraction of query segments whose ground-truth counterpart is found within the top $k$ ranked candidates, thus gauging how effectively the method surfaces correct matches among its highest-confidence predictions. Additional details for the dataset, baseline and experiments can be found in the supplementary, along with more qualitative results.

## 5 Results

**Segment Matching** In Table 1, we compare our proposed method SegMASt3R with the state-of-the-art methods using two categories of approaches in the literature: the well-established local feature matching (LFM) based on sparse or dense keypoint correspondences, and the recently emerging open-set instance association based on segment matching (SegMatch). It can be observed that SegMASt3R outperforms all the baselines for all the pose bins with a huge margin. Amongst the LFM techniques, dense matchers (RoMa and MASt3R) outperform sparse matchers (SP-LG and GiM-DKM) on both the evaluation metrics for the task of segment matching. Notably, on the highly challenging wide-baseline settings, MASt3R outperforms other local matchers by a large margin. However, when the same backend features are aggregated at the segment-level, MASt3R's performance deteriorates significantly, e.g., AUPRC drops from 52.9 to 41.4 on the 90-135 pose bin. This uncovers an inherent limitation: *feature distinctiveness at the pixel level does not necessarily translate to the instance level*. Considering the SegMatch techniques which are not trained for the LFM task, it can be observed that DINOv2 and SegVLAD perform particulary well on R@5 metric, which aligns with their typical use for coarse retrieval [19, 14]. On the other hand, SAM2's two-frame video propagation only works well for narrow-baseline matching, which can be expected as its training set is mostly comprised of dynamic object tracking. Overall, these results show that all

| Type | Method | 0°–45° | | | 45°–90° | | | 90°–135° | | | 135°–180° | | |
|------|--------|--------|------|------|---------|------|------|----------|------|------|-----------|------|------|
| | | AUPRC | R@1 | R@5 | AUPRC | R@1 | R@5 | AUPRC | R@1 | R@5 | AUPRC | R@1 | R@5 |
| LFM | MASt3R [22] | 78.2 | 86.5 | 89.4 | 69.5 | 77.6 | 81.0 | 48.0 | 60.4 | 64.6 | 32.5 | 49.0 | 54.1 |
| SegMatch | MASt3R [22] | 52.2 | 57.5 | 81.2 | 39.1 | 51.0 | 78.6 | 23.6 | 45.9 | 77.2 | 17.2 | 43.8 | 75.7 |
| Ours | SEGMAST3R | 95.0 | 96.0 | 98.6 | 86.2 | 91.2 | 96.4 | 73.4 | 85.2 | 95.7 | 68.4 | 83.8 | 94.8 |

Table 2: Performance of selected methods across pose-bins on Replica [39]. Blue cells mark the best scores; Orange cells mark the second-best.

the baseline methods lack on at least one of the fronts: pixel- vs segment-level distinctiveness, recall vs precision, and narrow- vs wide-baseline robustness. Our proposed method SegMASt3R achieves all these desirable properties with high performance across the board by learning segment-level representations with the training objective of instance association. In Section 5, we provide qualitative results which emphasize the ability of our method to address the problem of perceptual instance aliasing and instance matching under extreme viewpoint shifts.

**Generalization** We assess our model's ability to generalize to new environments. First, in Table 2, we present results on a different indoor dataset, Replica, to test the generalization ability of SegMASt3R, which is trained only on ScanNet++. We compare SegMASt3R against the LFM and SegMatch versions of MASt3R, as these three methods closely resemble each other in terms of their network architecture (detailed comparisons are included in the supplementary). It can be observed that the performance patterns on Replica remain largely the same as that on ScanNet++, and SegMASt3R consistently outperforms the LFM and SegMatch versions of MASt3R across the board.

Furthermore, we test generalization to challenging outdoor scenes from the MapFree dataset [2], with results shown in Table 3. Since MapFree lacks instance-level ground truth, we use SAM2's video propagator on image sequences to generate a pseudo-ground truth for evaluation. Our indoor-trained model (SegMASt3R (SPP)) shows a regression in performance in comparison to DINOv2, highlighting the domain shift. However, this gap can be substantially closed by either re-training on MapFree data (SegMASt3R (MF)) or, even more simply by just recalibrating the single learnable dustbin parameter $\alpha$ using a grid-search over a small calibration set from the target domain (SegMASt3R (SPP, Dustbin MF)). This demonstrates the strong adaptability of our model's learned geometric features.

| Method | Train Set | Overall IoU | 0-45° | 45-90° | 90-135° | 135-180° |
|--------|-----------|-------------|-------|--------|---------|----------|
| DINOv2 [30] | Multiple | 84.4 | 85.4 | 85.2 | 83.5 | 83.8 |
| MASt3R (Vanilla) [22] | Multiple | 69.2 | 73.4 | 69.8 | 70.0 | 66.1 |
| SegMASt3R (SPP) | ScanNet++ | 75.2 | 75.2 | 74.6 | 76.5 | 74.5 |
| SegMASt3R (SPP, Dustbin MF) | ScanNet++ | 88.7 | 88.6 | 88.6 | 88.5 | 88.9 |
| SegMASt3R (MF) | MapFree | 93.7 | 93.3 | 93.7 | 93.9 | 93.9 |

Table 3: Generalization performance on the outdoor MapFree dataset [2].

| Method | office0 | office1 | office2 | office3 | office4 | room0 | room1 | room2 |
|--------|---------|---------|---------|---------|---------|-------|-------|-------|
| | AP / AP@50 | AP / AP@50 | AP / AP@50 | AP / AP@50 | AP / AP@50 | AP / AP@50 | AP / AP@50 | AP / AP@50 |
| ConceptGraphs (MobileSAM Masks) [17] | 11.84 / 28.43 | 20.31 / 43.79 | 8.63 / 22.82 | 8.07 / 22.83 | 9.46 / 24.73 | 12.23 / 34.34 | 5.83 / 12.96 | 7.83 / 23.82 |
| ConceptGraphs (GT Masks) [17] | 43.53 / 69.68 | 22.48 / 40.71 | 43.46 / 60.69 | 32.06 / 53.44 | 39.63 / 68.22 | 44.89 / 69.64 | 17.96 / 36.53 | 25.93 / 43.63 |
| SegMASt3R(Ours, GT Masks) | 79.93 / 87.17 | 54.89 / 64.42 | 64.00 / 85.50 | 58.02 / 79.93 | 67.48 / 85.01 | 71.02 / 91.22 | 64.09 / 85.50 | 56.35 / 76.66 |

Table 4: Class-Agnostic instance-mapping performance (AP and AP@50) on Replica scenes, shown in percentage. The best value in each column is highlighted in blue.

**3D Instance Mapping** The goal of *instance mapping* is to localize object instances in 3D so a robot can distinguish objects of the same class in both image and metric space [17, 28, 44]. The main difficulty is preserving identities over long trajectories, especially when objects leave the camera's view and later reappear from different angles. Our pipeline employs SEGMAST3R to match object masks across image pairs. Given ground-truth masks and sampled pairs, we extract mask features, solve a Sinkhorn assignment, and take the row-wise argmax to obtain tentative correspondences.

| Method | IoU > 0.25 | | | IoU > 0.50 | | | IoU > 0.75 | | |
|---|---|---|---|---|---|---|---|---|---|
| | AUPRC | R@1 | R@5 | AUPRC | R@1 | R@5 | AUPRC | R@1 | R@5 |
| SAM2 [34] | 47.4 | 56.2 | 61.2 | 50.2 | 58.7 | 62.6 | 55.6 | 64.1 | 66.6 |
| DINOv2 [30] | 39.0 | 60.0 | 89.2 | 44.4 | 65.0 | 92.0 | 54.9 | 73.1 | 94.0 |
| MASt3R (Vanilla) [22] | 50.7 | 58.0 | 76.2 | 52.4 | 59.5 | 77.0 | 49.6 | 57.4 | 76.3 |
| SegMASt3R (Ours) | 84.2 | 91.9 | 99.3 | 87.6 | 94.4 | 99.3 | 89.9 | 94.3 | 96.3 |

Table 5: Performance on ScanNet++ [47] using noisy masks from FastSAM.

We then back-project each matched mask into 3D and drop links whose point-cloud IoU falls below 0.5, rejecting any match that fails this geometric check. Details are provided in the supplementary. Table 4 shows percentage AP versus ConceptGraphs [17]. We evaluate under two conditions: using masks generated by MobileSAM [48] as in the original ConceptGraphs setup, and using ground-truth (GT) masks for a fairer comparison of the underlying matching capability. Our geometry-aware matching yields higher accuracy in both settings, particularly when objects exit and later re-enter the field of view, demonstrating robustness to both noisy masks and challenging viewpoints.

**Robustness to Noisy Segmentation Masks**   To assess the practical viability of our approach in real-world scenarios where ground-truth masks are unavailable, we evaluated all methods on Scan-Net++ using imperfect segmentations generated by FastSAM. A key challenge in this setting is that evaluation can conflate the performance of the segment matcher with the quality of the upstream Automatic Mask Generator (AMG). A low score may not distinguish between a matching failure and an AMG failure where predicted masks do not align with any ground-truth segment.

To decouple these factors and isolate the core matching performance, we adopt an evaluation protocol that does not penalize methods for AMG errors. Specifically, while all methods take noisy masks as input, the evaluation is performed only on the subset of ground-truth correspondences for which a valid match between predicted segments was possible. As shown in Table 5, SegMASt3R maintains a substantial performance margin over all baselines across different evaluation thresholds. This result confirms that the strong geometric priors learned by our method provide significant robustness, enabling superior matching even when conditioned on inconsistent and noisy segment inputs.

**Object-level Topological Navigation**   Recent works [17, 15, 16] have explored the use of SAM2's open-set, semantically-meaningful instance segmentation for topological mapping and navigation. These methods rely on accurate segment-level association. To show the benefits of our proposed method on complex downstream tasks such as navigation, we considered an object topology-based mapping and navigation method, RoboHop [15], and swapped its SuperPoint+LightGlue based segment matching with our proposed segment matcher. This segment matching is used by RoboHop's localizer to match each of the object instances in the query image with those in the sub-map images, which is then used to estimate the currently-viewed object's path to the goal and correspondingly obtain a control signal. We used the val set of the ImageInstanceNav [21] dataset, which comprises real-world indoor scenes from HM3Dv0.2 [33]. We followed [32] for creating map trajectories and evaluating navigation performance.

In Table 6, we report navigation success for (vanilla) RoboHop and an enhanced version of it that uses SegMASt3R for segment association for localization and navigation. We use two metrics: SPL (Success weighted by Path Length) [1] and SSPL (soft SPL) [8]. Furthermore, we considered four different evaluation settings using two parameters that define the submap used for localizing the query image segments: *Submap Span* ($S_s$), which defines the total number of images sampled from the map based on the distance from the robot's current position, and *Submap Density* ($S_\rho$), which defines a subsampling factor to uniformly skip map images. These parameter configurations aid in testing narrow- and wide-baseline matching as well as the ability to avoid false positives. Table 6 shows that SegMASt3R consistently outperforms vanilla RoboHop. In particular, we achieve an absolute improvement of 27% SPL on one of the hardest parameter settings: $S_s = 16, S_\rho = 0.25$, where only 4 submap images are sampled due to a low density value. This shows that even with a very sparse submap, it is possible to maintain high navigation success rate, thus avoiding the typical trade-off between compute time and accuracy for the localizer. In Figure 2, we present a qualitative comparison between vanilla RoboHop's segment matching and that based on SegMASt3R: (left) the mismatch between the wall (orange) and the vanity cabinet for the vanilla methods leads to an incor-

| Method | $S_s = 16, S_\rho = 0.25$ | | $S_s = 16, S_\rho = 0.5$ | | $S_s = 32, S_\rho = 0.25$ | | $S_s = 32, S_\rho = 0.5$ | |
|---|---|---|---|---|---|---|---|---|
| | SPL | SSPL | SPL | SSPL | SPL | SSPL | SPL | SSPL |
| RoboHop [15] | 36.34 | 54.25 | 54.51 | 69.98 | 60.57 | 68.29 | 57.52 | 68.47 |
| SegMASt3R (Ours) | **63.60** | **78.84** | **63.60** | **78.33** | **66.62** | **75.20** | **63.56** | **73.89** |

Table 6: Navigation performance comparison. The best value in each column is highlighted in blue.

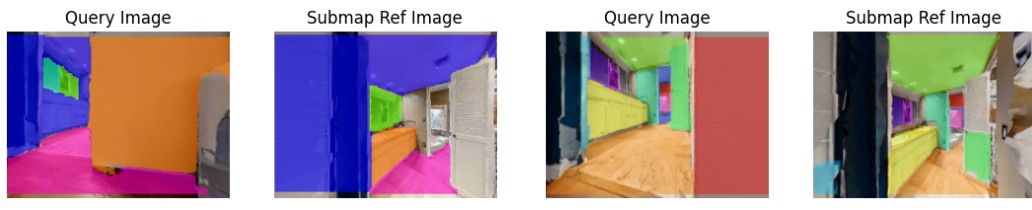

RoboHop w. DINOv2       w. SegMASt3R

Figure 2: **Segment Matching-Guided Navigation.** (left) In vanilla RoboHop's segment matching, a wall segment (orange) gets mismatched with the vanity cabinet and misguides the agent to move towards its right, leading to a navigation failure. (right) SegMASt3R correctly recognizes the same cabinet as well as other segments and guides the robot into the bathroom, and eventually to the goal. Note that the query and submap images vary across both the methods, as we manually probed the point of failure for the baseline and the nearest agent state for ours.

rect rotation towards the right, consequently leading to navigation failure, whereas SegMASt3R's accurate segment matching (right) guides the agent into the bathroom and finally to the goal object.

**Impact of the Feature Encoder** To isolate the contribution of the feature encoder in our pipeline, we replace MASt3R's cross–attention block with two alternative, purely 2D backbones: *CroCo* (shared by both MASt3R and SegMASt3R) and *DINOv2*, a state-of-the-art encoder often used for off-the-shelf segment matching [17, 15, 14]. As reported in Table 7, neither backbone yields competitive segment-matching accuracy. This result underscores that learning segment descriptors from a 2D foundation model alone is inadequate; geometric context is essential. In contrast, MASt3R's cross–attention layers, 3D-aware training regimen, and explicit formulation of image matching in 3D jointly endow the model with the priors required for reliable instance association. The fact that CroCo is always second best suggests that cross-view-completion, by itself, can yield superior results for segment matching between image pairs.

| Method | 0°–45° | | | 45°–90° | | | 90°–135° | | | 135°–180° | | |
|---|---|---|---|---|---|---|---|---|---|---|---|---|
| | AUPRC | R@1 | R@5 | AUPRC | R@1 | R@5 | AUPRC | R@1 | R@5 | AUPRC | R@1 | R@5 |
| DINOv2-SegFeat | 64.7 | 71.5 | 89.3 | 55.7 | 65.9 | 87.3 | 45.4 | 59.1 | 84.4 | 36.8 | 53.4 | 81.3 |
| CroCo-SegFeat | 73.4 | 78.8 | 92.3 | 64.0 | 73.1 | 90.6 | 50.7 | 64.6 | 87.5 | 38.5 | 56.6 | 84.1 |
| SegMASt3R (Ours) | 92.8 | 93.6 | 98.0 | 91.1 | 92.2 | 97.6 | 88.0 | 89.5 | 96.8 | 83.6 | 85.9 | 95.9 |

Table 7: Proposed method and model ablations performance across pose-bins on ScanNet++ [47]. Blue cells mark the best scores; Orange cells mark the second-best.

**Qualitative Results** Figure 4 compares SegMASt3R with SAM2's two-frame video propagation-based segment matching under *extreme viewpoint variations* on the ScanNet++ dataset. Each row shows a reference image, SAM2's matches, SegMASt3R's matches, respectively. In the top row, a wall (pink) and a door (blue) are mismatched by SAM2, whereas SegMASt3R correctly associates them, despite a very limited visual overlap. In the bottom row, SAM2 gets confused between the two monitors, whereas SegMASt3R is able to correctly associate them despite the simultaneous effect of 180° viewpoint shift and *perceptual instance aliasing* (i.e., different instances of the same object category in an image potentially lead to mismatches). Figure 3 presents qualitative results on the (outdoor) MapFree [2] dataset, where we compare our ScanNet-trained SegMASt3R with the off-the-shelf DINOv2 [30] features. Each image triplet represents the query segment (left) and its

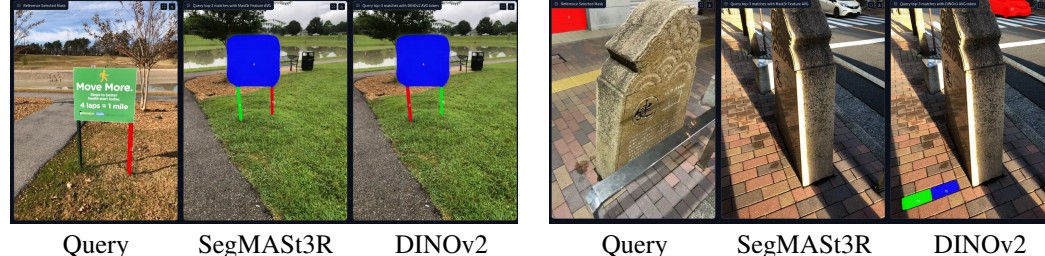

| Query | SegMASt3R | DINOv2 | Query | SegMASt3R | DINOv2 |

Figure 3: *MapFree Outdoor Dataset* - **Perceptual Instance Aliasing** (left): the right leg of the signboard as a query segment (red) is correctly matched by our method but mismatched with its left leg by DINOv2. **Sinkhorn Matches to Dustbin** (right): the query segment (red) is not visible in the reference image and is correctly ignored by our method, whereas DINOv2 mismatches it with a vehicle segment.

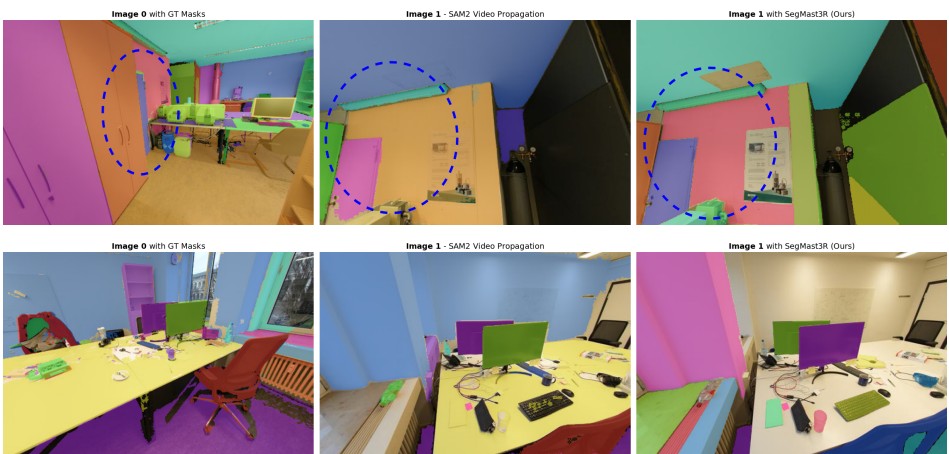

Figure 4: *ScanNet++ Dataset* – **Wide-baseline Matching** (top): The wall (pink) and the door (blue) in the query image (left) gets incorrectly associated by SAM2's video propagation (middle), whereas SegMASt3R (right) is able to correctly match them despite very limited visual overlap. **Perceptual Instance Aliasing** (bottom): unlike SAM2, SegMASt3R is able to correctly associate the pair of monitors under the simultaneous duress of an opposing viewpoint observation and perceptual instance aliasing.

retrieved matches using our method (middle) and DINOv2 (right). The query segment is displayed in red color, and its top three matches are respectively displayed in red, green, and blue colors. The left image triplet illustrates another case of perceptual instance aliasing. SegMASt3R is able to resolve this aliasing problem, whereas DINOv2 confuses the right and left legs of the signboard. The right image panel shows that our Sinkhorn solver effectively learns dustbin allocations for explicitly rejecting negatives, that is, the segments which do not have any corresponding match, whereas DINOv2 leads to incorrect matches.

# 6   Conclusion

We proposed SegMASt3R, a simple method to re-purpose an existing 3D foundation model MASt3R for image segment matching. Our proposed method achieves excellent results on ScanNet++ and Replica with a simple pipeline and a minimal amount of training. It especially excels on wide-baseline segment matching between image pairs. In addition, we show that SegMASt3R has practical applicability by evaluating it's performance on the downstream tasks of 3D instance mapping and object-relative topological navigation, where we significantly outperform the corresponding baselines. Overall, this work attempts to establish segment matching as a core computer vision capability, which will enable even more downstream applications in future alongside the advances in image segmentation and data association.

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
