# OpenReview forum: "SegMASt3R: Geometry Grounded Segment Matching"
_NeurIPS.cc/2025/Conference — NeurIPS 2025 spotlight_

### Official Review · Reviewer_nAMx · 2025-06-14

**Clarity:** 4
**Significance:** 3
**Originality:** 3
**Rating:** 5
**Confidence:** 4

**Summary:**

This paper presents SegMASt3R, a novel framework that adapts the 3D foundation model MASt3R for wide-baseline segment matching. It introduces a lightweight segment-feature head to generate segment-level descriptors and applies a differentiable optimal transport solver with a learnable dustbin. Extensive experiments demonstrate that SegMASt3R significantly outperforms both local feature matchers and 2D segment-based methods. Its effectiveness is further validated in downstream tasks such as 3D instance mapping and topological navigation.

**Questions:**

My concerns mainly lie in the evaluation settings. Please refer to the weaknesses.

**Ethical Concerns:**

["NO or VERY MINOR ethics concerns only"]

**Final Justification:**

The authors have addressed my concerns. I think the proposed SegMASt3R provides an effective solution for segment matching, and its potential has been demonstrated through downstream applications. Therefore, I maintain my rating of accept.

**Limitations:**

Yes, in the supplementary material.

**Paper Formatting Concerns:**

None.

**Quality:**

4

**Strengths And Weaknesses:**

Strengths:
1. Effective model design. The method builds upon a strong backbone (MASt3R). The design of the segment-level feature head and the differentiable matching mechanism with a learnable dustbin is reasonable.
2. Strong performance and diverse downstream applications. The application of SegMASt3R in topological navigation is interesting and demonstrates its practical value.
3. Clear writing. The paper is well-written and easy to follow.

Weaknesses:
1. Limited ablation:

The method uses a learnable dustbin to handle unmatched segments. It would strengthen the paper to include an ablation study evaluating its contribution.

2. Evaluation settings.

(1) It is unclear whether the baseline methods are re-trained or evaluated using their publicly released checkpoints. This is especially critical in wide-baseline settings, where fine-tuning or data augmentation strategies may significantly affect performance.

(2) According to Supplementary Table 2, SegMASt3R is evaluated only using ground-truth masks, which are often unavailable during inference. It would be beneficial to also report results using masks from MobileSAM.

3. Minor typo.

In Line 111, the channel dimension of V1, V2 should be 768, instead of 24.

---

> ### Author Rebuttal · Authors · 2025-07-31
>
> ### Limited ablation:
> >The method uses a learnable dustbin to handle unmatched segments. It would strengthen the paper to include an ablation study evaluating its contribution.
>
> The table below shows the effect of removing the dustbin parameter from the SegMASt3R pipeline. The performance sharply degrades across all metrics without the dustbin parameter.
>
> ### Ablation Study on Learnable Dustbin
>
> | Method                                             | 0–45 AUPRC | 0–45 R@1 | 0–45 R@5 | 45–90 AUPRC | 45–90 R@1 | 45–90 R@5 | 90–135 AUPRC | 90–135 R@1 | 90–135 R@5 | 135–180 AUPRC | 135–180 R@1 | 135–180 R@5 |
> |---------------------------------------------------|------------|----------|----------|-------------|-----------|-----------|----------------|-------------|-------------|------------------|---------------|---------------|
> | SegMASt3R (No Dustbin) | 65.0       | 73.8     | 95.9     | 61.8        | 71.5      | 95.4      | 58.7           | 69.1        | 94.6        | 55.1             | 66.4          | 93.6          |
> | **SegMASt3R**                                     | **92.8**   | **93.6** | **98.0** | **91.1**    | **92.2**  | **97.6**  | **88.0**       | **89.5**    | **96.8**    | **83.6**         | **85.9**      | **95.9**      |
>
> ### MobileSAM Mask Evaluation
> > According to Supplementary Table 2, SegMASt3R is evaluated only using ground-truth masks, which are often unavailable during inference. It would be beneficial to also report results using masks from MobileSAM.
>
> During inference, we can rely on segment generators like MobileSAM. In Reviewer **jith**'s rebuttal, we show results on  evaluation using noisy masks as obtained from FastSAM in the "*Performance under noisy masks*" section. Additionally, in our robotic navigation task (L253-290) we use masks generated from FastSAM on the HM3D dataset (Table 4). This clearly shows that our method is not reliant on ground truth masks. We will mention this more explicitly in the paper.
>
>
> ### Were Pre-trained Checkpoints Used
> >  It is unclear whether the baseline methods are re-trained or evaluated using their publicly released checkpoints. This is especially critical in wide-baseline settings, where fine-tuning or data augmentation strategies may significantly affect performance.
>
> Yes, we use pre-trained checkpoints. This is indeed a missing detail on our part and we will add the pre-trained checkpoints used for evaluation in the paper. For DINOv2, we use the DINOv2 ViT-G pre-trained checkpoint. For CroCo, we use the ViT-L pre-trained checkpoint provided in their repo. For other models (SAM2, MASt3R, etc.), we use the official checkpoints available in their respective repos.
>
>
> ### Typos
> > In Line 111, the channel dimension of V1, V2 should be 768, instead of 24.
>
>  Thank you for pointing this out. We will fix the typo.

---

> > ### Comment · Reviewer_nAMx · 2025-08-03
> >
> > Thanks to the authors for their rebuttal. I have read all the reviewers' questions and the response. I believe the authors have adequately addressed the concerns, particularly those about the generalization of SegMASt3R and mask generation. I tend to accept this paper.

---

> > > ### Author Response · Authors · 2025-08-04
> > >
> > > We thank Reviewer **nAMx** for engaging with our work and for their response to the rebuttal. We’re glad that the clarifications through our rebuttal addressed their concerns. We appreciate their support and positive assessment of our submission.

---

### Official Review · Reviewer_vtN9 · 2025-06-20

**Clarity:** 4
**Significance:** 4
**Originality:** 2
**Rating:** 6
**Confidence:** 4

**Summary:**

This paper addresses the challenging problem of segment-level image correspondence under extreme viewpoint variations, leveraging the spatial reasoning capabilities of 3D foundation models. The proposed method, SegMASt3R, utilizes a frozen MASt3R backbone to extract geometry-aware patch-level features from an image pair. These features are upsampled and aggregated into segment-level descriptors via a lightweight MLP, using segmentation masks from external models (e.g., SAM2) or ground truth. Segment correspondences are then established through a differentiable optimal transport layer based on the Sinkhorn algorithm. The model is trained end-to-end on ScanNet++ using a SuperGlue-inspired contrastive loss that accounts for both matched and unmatched segments. SegMASt3R is evaluated on ScanNet++ and Replica, showing strong generalization across indoor scenes, and is qualitatively assessed on the challenging MapFree outdoor dataset. It is further validated on downstream tasks such as 3D instance mapping and object-goal navigation, where it delivers substantial improvements over baselines. The method achieves up to a 30% gain in AUPRC over both segment-matching and local-feature-based methods.

**Questions:**

- The rebuttal could elaborate on the robustness to segmentation noise
- The rebuttal could elaborate on the dependence of performance on the number of segments
- The final version of the paper could include already in the main paper an abridged version of the discussion on limitations and societal impact

**Ethical Concerns:**

["NO or VERY MINOR ethics concerns only"]

**Final Justification:**

The authors have addressed all of my questions satisfactorily. After reviewing the other reports, I believe they engaged constructively in the review process and responded thoroughly to all concerns. I therefore confirm my recommendation to accept this paper.

**Limitations:**

Yes

**Paper Formatting Concerns:**

No concern about formatting

**Quality:**

4

**Strengths And Weaknesses:**

The main strength of this paper is the strong performance on a widely relevant computer vision task, and strong results (>80) even for extreme viewpoint changes. This work undoubtedly has applications in practical tasks related to robotic navigation and instance mapping, as also show in the ablation and supplementary experiments. And the pipeline used is relatively simple, while very effective. Finally, the paper is very well structured and written, with a strong and exhaustive experimental section and an excellent methodology section, which discusses in clear details all aspects of the pipeline.

At the same time, the simplicity of the proposed architecture leads to limited novelty: it is a straightforward adaptation of MASt3R with an added head comprising an MLP and Sinkhorn-based matching. Additionally, as noted by the paper's limitation section (included in the supplementary material, not in the main paper), the experiments do not cover fully generalisation to outdoor settings, and training requires ground truth 3D instance segmentation, which is not always easy to obtain. An additional limitation not covered by the paper is how robust the method is to noisy segmentation masks. Finally, while the matching strategy is efficient enough (0.579s forward pass at inference) it is not discuss how scalable the approach is; the paper mentions using 100 segments per image to be 'sufficient to cover the entire image while keeping computation affordable', but this leaves space for a discussion on how performance would change adding more segments.

Despite the weaknesses, this paper still provides a very strong contribution to the computer vision community, and the weaknesses are mitigated by the fact that in most practical applications 100 segments is a reasonable number (e.g., for SAM), and the strong performances with a simple framework are a positive.

---

> ### Author Rebuttal · Authors · 2025-07-31
>
> ### Novelty
> > it is a straightforward adaptation of MASt3R with an added head comprising an MLP and Sinkhorn-based matching.
>
> While some individual components used in our method are known within the community, we demonstrate that even these simple elements when integrated algorithmically can achieve outstanding results across a range of vision tasks, including wide-baseline segment matching, robotic navigation and instance segmentation. Crucially, we integrate these components in a novel and non-trivial configuration, which is key to the performance gains observed. Our approach also serves as an effective exploration of segment-based representations, which, despite the simplicity of our methodology, is shown to enhance performance in multiple robotic-vision settings showcasing the usefulness of segments as a useful intermediate-level scene representation. To our knowledge, this still remains under-explored. Furthermore, the introduction of a segment matching task and wide-baseline evaluation adds an additional layer of novelty, pushing the boundaries of current approaches in this domain.
>
> ### 3D Instance Segmentation
> > Training requires ground truth 3D instance segmentation, which is not always easy to obtain:
>
> Our method relies more on 2D instance masks than 3D. The benefit with 3D instance masks is they are more boundary aware. 2D instance masks can be easily generated to both train the network as well as use the network to evaluate on novel scenes like shown in the quantitative MapFree results of **jith's** rebuttal. Our method, even when trained only on indoor datasets generalizes better than compared to baselines on the outdoor MapFree.
>
>
>
> ### Noisy Masks
> > An additional limitation not covered by the paper is how robust the method is to noisy segmentation masks.
>
> In reviewer **jith**’s rebuttal we show results for the MapFree dataset in the Generalization / Outdoor Dataset section which covers both outdoors and noisy masks as the masks generated are via SAM2. Additionally, we show results on noisy masks as generated by FastSAM in  **jith**'s  in the *Performance under noisy masks* section. The table shows that the performance patterns are still similar to that observed in Table 1 with GT masks.
>
> ### Dependence of performance on the number of segments:
> > the paper mentions using 100 segments per image to be 'sufficient to cover the entire image while keeping computation affordable, but this leaves space for a discussion on how performance would change adding more segments.
>
> Our method is not reliant on a particular number of segments. The M (set to 100) parameter in L117 is an upper bound on the segments in the scene. It allows for more efficient batch processing. For the missing masks, we just pad them with 0s. We will add this important detail to the paper (in Line 117).  Additionally, during inference time, the number of masks can be set arbitrarily regardless of the set M during training. We presume beyond a certain point having really small segments would indeed degrade performance, but most segmenters (like SAM2, FastSAM) yield less than a 100 segments. Additionally, in the main paper, and in the rebuttal, we have explored several results with different segmentation algorithms. FastSAM is used in the robotic navigation experiment (Table 4 on the HM3D dataset), SAM2 AMG (Automatic Mask Generator) results are shown for the *Performance under noisy masks* section for reviewer **jith**. The various segmenters used also naturally cover a varying number of segments, showing that indeed our methods are robust to this quantity.

---

> > ### Comment · Reviewer_vtN9 · 2025-08-01
> >
> > I thank the authors for their thorough and thoughtful response. The rebuttal is detailed, and all of my concerns have been satisfactorily addressed.
> >
> > In particular, the issues regarding robustness to noise and potential limitations related to the number of segments have been clarified with additional evaluations, which demonstrate how SegMASt3R outperforms current state-of-the-art methods.
> >
> > The concern about the limited novelty - specifically in terms of combining existing modules - has also been addressed. While the theoretical innovation remains limited, the substantial performance gains achieved through this integration underscore the practical relevance and impact of the work for the community.

---

> ### Author Response · Authors · 2025-08-04
>
> We thank Reviewer **vtn9** for taking the time to review our paper and responding to our rebuttal. We’re glad we were able to address their concerns.

---

### Official Review · Reviewer_mKs5 · 2025-07-01

**Clarity:** 3
**Significance:** 3
**Originality:** 2
**Rating:** 4
**Confidence:** 4

**Summary:**

SegMASt3R is a model for segment matching, an under-explored problem. That is, the paper introduces the wide baseline segment matching problem between image pairs. An approach based on frozen MASt3R feature extractor, given objects masks, and the differentiable optimal transport framework of SuperGlue is provided. In addition, a benchmark for wide segment matching is proposed with
quantitative evaluations on indoor datasets ScanNet++ and Replica and show visual results on MapFree Outdoor dataset. The proposed approach improves downstream tasks such as 3D instance mapping and object level topological navigation.

**Questions:**

Q1. Why downsample the masks for the DINOv2 and MASt3R baselines, rather than interpolating their features to match the
mask resolution?
Q2. The masks extraction is a crucial step in creating high quality segments. ScanNet++ and Replica dataset provide masks, however, in cases where masks are not provided annotating 100 masks (line
123) for every input image seems unpractical. When ground truth
masks are not available, how do the authors propose to obtain it? What
is the expected running time for this step?
Q3. Have the authors considered comparing to "Gaussian Grouping: Segment and Edit Anything in 3D Scenes"?

-Q4: Statistical Error Bars: Question 7 checklist: I don’t think it is correct – there is no
statistical error bars in any of the results (at least not mentioned) also in 3.4.1 the
time is not with standard deviation.

**Ethical Concerns:**

["NO or VERY MINOR ethics concerns only"]

**Final Justification:**

I thanks the authors regarding the clarification about the 3d annotation. While I'm still not convinced about the novelty, in light of this clarification I am raising the score from Borderline Reject to Borderline Accept.

**Limitations:**

The paper overall limitation is that it requires a 3D ground truth annotation, which are quite
rare. Furthermore, as a direct result of that, they cannot generalize well to other scenes and
datasets. Those issues were addressed by the authors in the Sup. Material.

**Quality:**

2

**Strengths And Weaknesses:**

Strengths:
- An approach for the under explored problem of wide baseline segment matching is offered, showing improved results over existing segment matching methods on indoor datasets.
- Using SegMASt3R improves performance on downstream tasks such as 3D instance mapping and Object level topological navigation
-  The paper is, overall, well written, the math is clearly explained, and the description of the
figures is clear.

Weaknesses:
- All quantitative results are reported exclusively on indoor datasets.
1. ScanNet++ , an indoor dataset, which the proposed method is
trained on although the competitor&#39;s methods aren&#39;t. 2. Replica, an
indoor synthetic dataset.
- The comparison with DINOv2 and MAStr3 methods in table 1 appears
to favor the proposed approach. The proposed method first upsample
the features to the image resolution before the mask are applied (lines
114-115), however, for extracting segment match from dense features
encoders the authors downsample the masks to the features&#39;
degraded resolution (lines 199-202).
-  The proposed approach mainly combines existing components without
introducing substantial methodological advances.


Minor weaknesses:
- Inaccuracy in the abstract. Lines 9-12 &quot;Extensive experiments show
that our approach outperforms state-of-the-art methods, including
the SAM2video propagator and local feature matching methods, by up
to 30% on the AUPRC metric, on ScanNet++ and Replica datasets.&quot;;
Since many of those methods were not inherently built for the task of
segment matching some required modifications by the authors to
produce segment correspondences. Therefore, it is not accurate to
name those methods SOTA for this task or to claim the proposed
method outperform SOTA methods.
- The necessity for the learnable logit in the learnable dustbin is unclear.

---

> ### Author Rebuttal · Authors · 2025-07-31
>
> ### DinoV2 and MASt3R Feature Resolution
> > Q1. Why downsample the masks for the DINOv2 and MASt3R baselines, rather than interpolating their features to match the mask resolution?
>
> We only downsample the masks for DINOv2, not for MASt3R (we will rectify this in L202). For DINOv2, the following table compares different feature/mask resolutions. It can be observed that performance does not change significantly. We had originally tested this on a subset and chose a lower resolution ([112, 171] - one-third of the original resolution) for efficiency.
>
> | Resolution ($[H,W]$)                                                       | 0–45 AUPRC | 0–45 R@1 | 0–45 R@5 | 45–90 AUPRC | 45–90 R@1 | 45–90 R@5 | 90–135 AUPRC | 90–135 R@1 | 90–135 R@5 | 135–180 AUPRC | 135–180 R@1 | 135–180 R@5 |
> |--------------------------------------------------------------|------------|----------|----------|-------------|-----------|-----------|----------------|-------------|-------------|------------------|---------------|---------------|
> | $[112,171]$       | **0.5794** | **0.6670** | **0.8742** | **0.4300**  | 0.5586    | 0.8319    | **0.3347**     | **0.4804**  | **0.7797**  | **0.3239**       | **0.4602**    | **0.7563**    |
> | $[168,256]$                 |   0.5784   | 0.6663    | 0.8740    |  0.4297     | 0.5584 | 0.8315 |  0.3323 |  0.4801 | 0.7794 | 0.3234 | 0.4597 | 0.7498 |
> | $[336,512]$                 | 0.5788     | 0.6657   | 0.8739   | 0.4297      | **0.5589**| **0.8330**| 0.3312         | 0.4803      | 0.7797      | 0.3203           | 0.4600        | 0.7563        |
>
> ### Number of Masks, Non-GT Masks For Training, Running Time for Mask Generation At Inference
>
> > Q2. The masks extraction is a crucial step in creating high quality segments. ScanNet++ and Replica dataset provide masks, however, in cases where masks are not provided annotating 100 masks (line 123) for every input image seems unpractical. When ground truth masks are not available, how do the authors propose to obtain it? What is the expected running time for this step?
>
> We respond to this in three parts: a) the choice of M=100, b) the need for ground truth (GT) masks during training, and c) expected runtime of non-GT masks during inference.
>
> ### Number of Masks
> a) **Number of Masks / M=100 masks**: This is simply an upper bound for convenience in batch processing. Most training/test pairs do not have 100 masks, regardless of using GT or SAM/FastSAM. The actual number of masks per image is between 20 to 30 when training using GT. For the missing masks, we just pad them with 0s. We will add this important detail to the paper (in Line 117). Additionally, during inference time, the number of masks can be set arbitrarily regardless of the set M during training.
>
> ### Non-GT Masks For Training
> b) **Need for GT masks during *training***: It is indeed possible to train SegMASt3R with *weak supervision* based on mask annotations obtained from SAM2 video propagator. To show this, we have now trained a model on the MapFree dataset, which does not have GT instance masks but has image sequences per scene. We use this with SAM2's video propagator (which works well due to narrow baseline between adjacent frames) to generate segment correspondences for different image pairs within the image sequence. This pseudo ground truth is used for both training and evaluation. For results on MapFree which involves noisy (non-GT masks) please refer to the **Generalization / Outdoor Datasets** section of reviewer **jith**'s.
>
> ### Running Time for Mask Generation At Inference
> c) Non-GT masks during *inference*:  When GT masks are unavailable, we can use SAM /FastSAM to obtain them. FastSAM works in real-time, so mask extraction should not be a bottleneck either in our training or inference and have real-time applicability.
>
> | **Method** |  **Overall IoU** |  **$0^\circ-45^\circ$** | **$45^\circ-90^\circ$** | **$90^\circ-135^\circ$** | **$135^\circ-180^\circ$** |
> | :--------: | :-------: | :-------: | :-------: | :-------: | :-------: |
> |DinoV2-SegFeat | 0.844 | 0.8543 | 0.8519 | 0.8351 | 0.8385 |
> |MASt3R (Vanilla) |  0.692 | 0.7336 | 0.6976 | 0.7001 | 0.6611 |
> |SegMASt3R (MF)  |  **0.937** | **0.9334** | **0.9374** | **0.9387** | **0.9398** |
>
>
> ### Gaussian Grouping
> > Q3. Have the authors considered comparing to "Gaussian Grouping: Segment and Edit Anything in 3D Scenes"?
>
> We thank the reviewer for pointing out Gaussian Grouping. We will cite the paper as it is related to this research, but can’t do a fair comparison, as our method is based on only *pairwise* image based segment matching. Gaussian Grouping on the other hand uses Gaussian splatting which requires *multiple overlapping views* of a scene to generate accurate segmentations.
>
> ### Statistical Error Bars
> > Q4: Statistical Error Bars: Question 7 checklist: I don’t think it is correct – there is no statistical error bars in any of the results (at least not mentioned) also in 3.4.1 the time is not with standard deviation.
>
> This was an oversight on our part. We meant to write no statistical error bars.
>
>
> ### Generalization to outdoor datasets
> > All quantitative results are reported exclusively on indoor datasets.
>
> Please refer to the *Generalization / Outdoor Dataset* section in reviewer **jith**'s rebuttal.
>
> >  comparison with DINOv2 and MAStr3 methods
>
> This is covered above in Q1.
>
> ### Novelty
> > The proposed approach mainly combines existing components without introducing substantial methodological advances.
>
> While some individual components used in our method are known within the community, we demonstrate that even these simple elements when integrated algorithmically can achieve outstanding results across a range of vision tasks, including wide-baseline segment matching, robotic navigation and instance segmentation. Crucially, we integrate these components in a novel and non-trivial configuration, which is key to the performance gains observed. Our approach also serves as an effective exploration of segment-based representations, which, despite the simplicity of our methodology, is shown to enhance performance in multiple robotic-vision settings showcasing the usefulness of segments as a useful intermediate-level scene representation. To our knowledge, this still remains under-explored. Furthermore, the introduction of a segment matching task and wide-baseline evaluation adds an additional layer of novelty, pushing the boundaries of current approaches in this domain.
>
> ### Abstract
> > Inaccuracy in the abstract
>
> We thank the reviewer for recognizing that we are introducing a new task (wide-baseline segment matching) and its evaluation methodology. To develop reasonable baselines, we chose to use SAM2 video propagator which is not inherently trained for this task, because it is the industry and academia standard for segment matching across multiple frames and is also trained on a large amount of data (more than MASt3R).
> We chose LFM methods because they have been repurposed as segment matchers (for example in [15,31]) and there is a straightforward methodology to turn them into segment matchers. It was our oversight that we only mentioned the baselines but did not explain their motivations, and will add these motivations to the main paper.
>
>
> ### Learnable Logit Interpretation and Learnability
> > The necessity for the learnable logit in the learnable dustbin is unclear.
>
> The logit is basically a threshold below which any score from the score matrix will be classified as not-matched. It is useful in situations when there are masks not common to both images and allows for such masks to be mapped to the dustbin. While we could also have let it have been a tuneable hyper-parameter, we chose to make it learnable to allow for a more natural discovery of its appropriate value. We will clarify this in the paper.
>
> ### Limitations
> > The paper overall limitation is that it requires a 3D ground truth annotation, which are quite rare. Furthermore, as a direct result of that, they cannot generalize well to other scenes and datasets. Those issues were addressed by the authors in the Sup. Material.
>
> We have now presented results on the MapFree dataset where we use SAM2 video propagator instead of ground truth annotations for training/evaluation. While this strategy may not apply for all types of datasets, it shows that our method does not *strictly* depend on ground truth annotations.

---

> > ### Comment · Reviewer_mKs5 · 2025-08-03
> >
> > I thank the authors for their response.
> > I keep my score as it, as I remained unconvinced regarding the novelty, nor about the need for 3d annotation.

---

> > > ### Author Response · Authors · 2025-08-04
> > >
> > > We thank mks5 for their detailed review, and a timely comment in this author-reviewer discussion phase.
> > >
> > > ## 3D Annotation
> > > 3D annotation is ***NOT*** a requirement for our method. Line 152 in the supplementary was mainly a pointer to the availability of 3D instance-level annotation in the ScanNet++ dataset. While we derive 2D instance associations from 3D annotations on ScanNet++ for training, we have now demonstrated a training pipeline using the MapFree dataset ***which neither requires object instances annotations nor 3D annotations*** (please refer to Section “Generalization / Outdoor Dataset” in **jith**’s rebuttal). This is achieved by using SAM2’s video propagation over multiple frames (with consecutive narrow baselines) to obtain segment correspondences across wider baselines for training.
> > >
> > >
> > > ## Novelty
> > > In addition to our response in the “Novelty” section above, we emphasize the novelty of our paper in terms of the **new knowledge/insights** that have never been reported in the prior art. We have established, for the first time, that “feature distinctiveness at the pixel level does not necessarily translate to the instance level” (L225). We show that a highly accurate local feature matching method (MASt3R) falls short at segment/instance matching when used off-the-shelf (Table 1). Furthermore, our ablation studies (Table 5 and 2nd table in **mKs5**’s rebuttal) show a surprising trend inversion in how different leading backbones behave: DINOv2 (a 2D foundation model) is better than MASt3R (a geometric foundation model) for off-the-shelf segment matching (SegMatch block in Table 1), but SegMASt3R is better than DINOv2-SegFeat when trained for the task of segment matching (Table 5), which reinforces our claim that “geometric context is essential” (L286) for the segment matching task.
> > >
> > > Finally, we would like to re-emphasize the simplicity of our proposed pipeline (which in itself never existed before) as a rather positive aspect of this work that is able to deliver remarkable results for the underexplored segment matching task as well as the downstream tasks of mapping and navigation.

---

> ### Author Response · Authors · 2025-08-08
>
> We note that our earlier response provided details addressing the reviewer’s concerns on novelty and the requirement of 3D annotation. If Reviewer **mKs5** has any remaining specific points, we are happy to clarify further.

---

### Official Review · Reviewer_jith · 2025-07-01

**Clarity:** 2
**Significance:** 3
**Originality:** 2
**Rating:** 4
**Confidence:** 4

**Summary:**

The paper introduces SegMASt3R, a segment matching framework that leverages a 3D foundation model (MASt3R) to establish wide-baseline segment correspondences. The key contribution lies in the segment-feature head, which aggregates patch-level features into segment-level descriptors, followed by a differentiable optimal transport layer for segment matching. The approach demonstrates strong performance improvements on ScanNet++ and Replica, supported by qualitative results on MapFree, and is further evaluated on downstream tasks such as 3D instance mapping and topological navigation.

**Questions:**

- The segmentation masks in this paper are from ground truth or SAM2. How does SegMASt3R perform when using imperfect or noisy segment masks?
- Have you conducted ablations on fine-tuning the cross-attention decoder in MASt3R jointly with the segment-feature head? Could this further enhance segment matching?

**Ethical Concerns:**

["NO or VERY MINOR ethics concerns only"]

**Final Justification:**

Thanks author for the detailed replies and additional experiments. My concerns are mostly solved, I'd like to raise my score to borderline accept.

**Limitations:**

Yes

**Quality:**

3

**Strengths And Weaknesses:**

- Strengths

  - Leverage 3D priors from pretrained model. SegMASt3R introduces MASt3R’s strong 3D inductive bias to address the gap in geometry-aware segment matching, enabling robust performance under wide viewpoint variations where 2D segment-based models may fail.

  - Strong experimental performance. The proposed method significantly outperforms existing baselines across all pose bins (Tab. 1, Tab. 2), especially in the wide-baseline setting (135–180°) in the ScanNet++ and Replica datasets.

  - Good downstream task application. The proposed method shows strong applicability to real-world tasks, including 3D instance mapping (Tab. 3) and object-based topological navigation (Tab. 4). These results highlight its ability to maintain object-level consistency across large viewpoint shifts.

- Weaknesses

    - Novelty concerns. While the paper leverage pretrained 3D foundation model as priors for segment matching is novel.  The core components are all adapted from prior work.
        -  The corss-view patch features are from MASt3R prediction.
        -  The core segment-feature head, is directly adpoted and initilized from MASt3R's descriptor head.
        - The differentiable matching layer and sinknorn normalization closely follow the SuperGlue approach.

     Hence, the contribution is largely engineering-driven.

    - Writing and Presentation Clarity.
        - Line 106: Patch-level features are of size (H/16, V/16, 768) → Should be (H/16, W/16, 768).
        - Line 99: “Cross-view encoding Transformer”. Encoding transformer or decoder?
        - Line 111: V1, V2 of shape (H/16, W/16, 24) → The patch-level embeddings V1, V2 should be (H/16, W/16, 768).
        - The learnable logit \alpha is not well defined and not used in the following equations. The description of $\alpha$ in line 152 is even confusing.
        - Lines 199–200: “A segment descriptor is the masked average of its feature map at a suitable resolution” — what is the suitable resolution? “Average of its feature map” — how is the average conducted?
        - Table 1: Local Feature Matching (LFM). Are the results from local visual descriptor matching or from segment matching?
        - Line 287: The ablation with CroCo encoder cannot demonstrate the effectiveness of the “explicit formulation of image matching”. To demonstrate it, it is better to use the DUSt3R backbone as it is not trained for the matching task.

    - Generalization concerns to outdoor. The evaluation on the outdoor MapFree dataset is limited to qualitative results, without quantitative metrics or comparisons. Including quantitative analysis, additional outdoor datasets, or more in-the-wild examples would strengthen the claim of generalization capability.

---

> ### Author Rebuttal · Authors · 2025-07-31
>
> **Typos:** We thank the reviewer for pointing out the typos.
>
> ### Cross-view Encoding Transformer
> > Line 99: “Cross-view encoding Transformer”. Encoding transformer or decoder?
>
> Thank you for pointing this out we will make the required change. It is indeed the decoder.
>
> ### Description of Dustbin
> >$\alpha$ Definition. Description of $\alpha$ in L152 is confusing:
>
> The learnable logit $\alpha$ is a threshold below which any score from the score matrix will be classified as not-matched [34]. It is useful in situations when there are no common masks between an image pair, as it then maps such masks to the dustbin. We will clarify this in the paper.
>
> ### Resolution and Averaging of Features
> > Lines 199–200: “A segment descriptor is the masked average of its feature map at a suitable resolution” - what is the suitable resolution? “Average of its feature map” - how is the average conducted?
>
> This is a valid point since we haven't explicitly documented the formulation in our paper. For DINOv2 specifically, we work with patch level features at 1/14th the image resolution $(H/14, W/14, 1536)$. To assign segment level features, we choose an intermediate resolution (1/3rd image resolution, decided empirically), and upsample the patch-level features to $(H/3, W/3, D)$ via bilinear interpolation. Binary segemnt masks are downsampled to this same resolution $(H/3, W/3, M)$ via nearest-neighbour interpolation. Finally to obtain our mask level features, we perform a mask-average pooling operation, implemented as a mat-mul: $(M, H/3 \times W/3) \times (H/3 \times W/3, D) = (M, D)$ to obtain a single D-dimensional vector per segment.
>
>
> ### Local Feature Matching for Evaluation
> > Table 1: Local Feature Matching (LFM). Are the results from local visual descriptor matching or from segment matching?
>
>  In this paper we believe we are the first to introduce wide-baseline segment matching. To develop a suitable baseline of methods, amongst others we chose to add LFM based baselines as they have been repurposed as segment matchers (for example in [15,31]) and there is a straightforward methodology to turn them into segment matchers. We will rectify this in the baselines subsection to clarify our motivation behind using them in the main paper.
>
> ### DUSt3R evaluation and fine-tuning MASt3R
> > Line 287: The ablation with CroCo encoder cannot demonstrate the effectiveness of the “explicit formulation of image matching”. To demonstrate it, it is better to use the DUSt3R backbone as it is not trained for the matching task.
> > Have you conducted ablations on fine-tuning the cross-attention decoder in MASt3R jointly with the segment-feature head? Could this further enhance segment matching?
>
> In Table 5 in the main paper and the Impact of the Feature Encoder (L281-L290) section, we were interested in performing an experiment to understand whether knowledge of the image pair and the subsequent cross-attention that features from both these images undergo is indeed a consequential part of these backbones for segment matching. As mentioned in the section, even though CroCo was not trained for the matching task and it was trained on much-less data than DINOv2, it still performs better than DINOv2 confirming our intuition. We explicitly did not choose DUSt3R because it is trained on a matching task and we imagined it would perform similar to MASt3R. We add the DUSt3R backbone comparison here for completion. As can be clearly seen SegMASt3R is much better than DUSt3R with the same training scheme as SegMASt3R on ScanNet++.  Additionally, we observe that fine-tuning the MASt3R decoder layers does not lead to further improvements in performance. Here we report by fine-tuning a total of 4 decoder layers (2 in each decoder).
>
> | Method                                             | 0–45 AUPRC | 0–45 R@1 | 0–45 R@5 | 45–90 AUPRC | 45–90 R@1 | 45–90 R@5 | 90–135 AUPRC | 90–135 R@1 | 90–135 R@5 | 135–180 AUPRC | 135–180 R@1 | 135–180 R@5 |
> |---------------------------------------------------|------------|----------|----------|-------------|-----------|-----------|----------------|-------------|-------------|------------------|---------------|---------------|
> | **SegMASt3R**                                     | **92.8**   | **93.6** | **98.0** | **91.1**    | **92.2**  | **97.6**  | **88.0**       | **89.5**    | **96.8**    | **83.6**         | **85.9**      | **95.9**      |
> | DUST3R-SegFeat                                    | 83.8       | 85.9     | 94.9     | 81.1        | 83.7      | 94.2      | 74.5           | 78.6        | 92.4        | 64.5             | 71.3          | 89.9          |
> | CroCo-SegFeat                                     | 73.4       | 78.8     | 92.3     | 64.0        | 73.1      | 90.6      | 50.7           | 64.6        | 87.5        | 38.5             | 56.6          | 84.1          |
> | SegMASt3R + Decoder Fine-Tuned                    | 90.3       | 91.5     | 96.7     | 88.6        | 90.1      | 96.3      | 84.9           | 87.1        | 95.2        | 78.8             | 82.5          | 93.7          |
>
> ### Generalization / Outdoor Dataset
> >The evaluation on the outdoor MapFree dataset is limited to qualitative results, without quantitative metrics or comparisons.
>
> Here's a quantitative comparison on the outdoor MapFree (MF) dataset using SAM2 masks. SegMASt3R (MF) is the model which we have now trained on the MapFree dataset (in response to Reviewer **jith**, **mKs5** comments). SegMASt3R (SPP) is the same model as that used in the original manuscript. SegMASt3R (SPP, Dustbin MF) differs from the previous only in the choice of dustbin parameter $\alpha$ which we borrowed from the SegMASt3R (MF) model to demonstrate that the indoor-to-outdoor generalization gap can be further reduced *easily* (e.g., this single parameter could simply be grid-searched on a small calibration set as an alternative to full training on MapFree).
>
> | **Method** | **Train Dataset** | **Eval Dataset** |  **Overall IoU** | IoU / **$0^\circ-45^\circ$** | IoU / **$45^\circ-90^\circ$** | IoU / **$90^\circ-135^\circ$** | IoU / **$135^\circ-180^\circ$** |
> | :--------: | :-------: | :--------: | :-------: | :-------: | :-------: | :-------: | :-------: |
> |DINOv2 | - | MapFree  | 84.4 | 85.4 | 85.2 | 83.5 | 83.8 |
> |MASt3R (Vanilla) | - | MapFree | 69.2 | 73.4 | 69.8 | 70.0 | 66.1 |
> |SegMASt3R (SPP)  | ScanNet++ | MapFree | 75.2 | 75.2 | 74.6 | 76.5 | 74.5 |
> |SegMASt3R (SPP, Dustbin MF)  | ScanNet++ | MapFree | 88.7 | 88.6 | 88.6 | 88.5 | 93.9 |
> |SegMASt3R (MF)  | MapFree | MapFree | 93.7 | 93.3 | 93.7 | 93.9 | 93.9 |
>
> **Ground Truth**: Most outdoor *real-world* datasets, including MapFree, don’t have ground truth instance masks, which is why we hadn't evaluated it quantitatively. The above evaluation uses a pseudo ground truth, obtained from SAM2 video propagation on the image sequence of a given scene. This works well in general because of the narrow baseline in adjacent video frames. We manually verified this ground truth on randomly selected scenes. All compared methods use the same pseudo ground truth.
>
> **Dataset/Training**: The official train split of the MapFree dataset comprises 460 unique scenes with 0.5 million images in total. To train SegMASt3R, we uniformly sampled 50 scenes to obtain 31K image pairs. Since the MapFree test split ground truth is not available publicly, we report results on their official val split using 7.8k image pairs from 13 scenes sampled uniformly from 65 total val scenes. Similar to our ScanNet++ evaluation, we use pose binning to present results across narrow and wide baselines.
>
> ### Performance under noisy masks
> > How does SegMASt3R perform when using imperfect or noisy segment masks?
>
> Our method is robust to noisy masks, e.g., those obtained from SAM2/FastSAM. This is evident from the use of SAM2 masks in the above results on the MapFree dataset, and FastSAM masks in the navigation results on the HM3D dataset (Table 4). The following table presents results on the ScanNet dataset using FastSAM, which shows that the performance patterns are still similar to that observed in Table 1 with GT masks:
>
> | Method            | 0-45 AUPRC | 0-45 R@1 | 0-45 R@5 | 45-90 AUPRC | 45-90 R@1 | 45-90 R@5 | 90-135 AUPRC | 90-135 R@1 | 90-135 R@5 | 135-180 AUPRC | 135-180 R@1 | 135-180 R@5 | Overall AUPRC | Overall R@1 | Overall R@5 |
> |-------------------|------------|----------|----------|-------------|-----------|-----------|---------------|-------------|-------------|----------------|--------------|--------------|----------------|-------------|-------------|
> | SAM2 (Video Prop) | 0.6389     | **0.7516**   | **0.7681**   | 0.5029      | **0.5884**    | 0.615     | 0.3142        | 0.3823      | 0.4175      | 0.2019         | 0.2593       | 0.2875       | 0.4147         | 0.4956      | 0.5222      |
> | MASt3R            | 0.3437     | 0.4014   | 0.5203   | 0.2862      | 0.3443    | 0.4575    | 0.2472        | 0.3067      | 0.4242      | 0.2370         | 0.2974       | 0.4220       | 0.2786         | 0.3375      | 0.4560      |
> | DinoV2-g14-AVG    | 0.3920     | 0.4563   | 0.6109   | 0.2596      | 0.3296    | 0.4883    | 0.1991        | 0.2701      | 0.4173      | 0.1737         | 0.2455       | 0.3784       | 0.2562         | 0.3254      | 0.4738      |
> | SegMASt3R AMG (Automatic Mask Generator)    | **0.7089**     | 0.7443   | 0.7694   | **0.6825**      | 0.7205    | **0.7455**    | **0.6444**        | **0.6883**      | **0.7178**      | **0.6032**         | **0.6548**       | **0.6827**       | **0.6598**         | **0.7020**      | **0.7289**      |
>
> ### Novelty
> > The core components are all adapted from prior work
>
> Please refer to our response to Reviewer **mKs5**, titled Novelty.

---

> > ### Comment · Reviewer_jith · 2025-08-03
> > **Thanks for the thorough response**
> >
> > I appreciate the authors for addressing my concerns regarding the additional DUSt3R backbone, generalizability to outdoor scenes, MASt3R backbone fine-tuning, and robustness to noisy masks. I may still have some concerns:
> >
> > Similar to other reviewers, I still have concerns about the novelty, as the key backbone difference only lies in using MASt3R’s descriptor head as a segmentation head for semantic matching.
> >
> > Additionally, I would like to point out that DUSt3R is not trained for the matching task. The key distinction between DUSt3R and MASt3R is the incorporation of a matching loss. Therefore, using the DUSt3R backbone as a comparison to MASt3R's could help demonstrate the effectiveness of the **explicit formulation of image matching**.
> >
> > In the additional noisy mask results, why not directly evaluate on ScanNet++ with noisy masks to compare with the results in Table 1?
> >
> > Best,
> >
> > jith

---

> ### Author Response · Authors · 2025-08-04
>
> Thanks very much, **jith**, for your response.
>
> ## Novelty
> In addition to our response in the “Novelty” section in the rebuttal, we emphasize the novelty of our paper in terms of the **new knowledge/insights** that have never been reported in the prior art. We have established, for the first time, that “feature distinctiveness at the pixel level does not necessarily translate to the instance level” (L225). We show that a highly accurate local feature matching method (MASt3R) falls short at segment/instance matching when used off-the-shelf (Table 1). Furthermore, our ablation studies (Table 5 and 2nd table in mks5’s rebuttal) show a surprising trend inversion in how different leading backbones behave: DINOv2 (a 2D foundation model) is better than MASt3R (a geometric foundation model) for off-the-shelf segment matching (SegMatch block in Table 1), but (Seg)MASt3R is better than DINOv2 when trained for the task of segment matching (Table 5), which reinforces our claim that “geometric context is essential” (L286) for the segment matching task.
> Finally, we would like to re-emphasize the simplicity of our proposed pipeline (which in itself never existed before) as a rather positive aspect of this work that is able to deliver remarkable results for the underexplored segment matching task as well as the downstream tasks of mapping and navigation.
>
> ## Backbone difference
> Additionally, we clarify that our segment-feature head is separate from MASt3R's local feature matching head (Line 113-115) and that we only initialize with local feature head’s weights. Since we keep the decoder frozen (Line 108), we retain MASt3R’s original local feature matching head, enabling simultaneous matching capability at both point and segment level through a single forward pass. Further, we have demonstrated that a separate segment-matching head is necessary since the MASt3R’s original local features are not amenable to segment-level aggregation/description, as observed in Table 1 (MASt3R in the SegMatch block).
>
> ## Noisy Masks on ScanNet++:
> In the Table below, we now present results on ScanNet++ using FastSAM masks for both the reference and the query image, which can be directly compared with Table 1. It can be observed that SegMASt3R achieves superior results across the board by a large margin, whereas even on the narrow baseline setting our framework is on par with the best of the methods being compared.
>
> | Method | AUPRC (0-45) | R@1 (0-45) | R@5 (0-45) | AUPRC (45-90) | R@1 (45-90) | R@5 (45-90) | AUPRC (90-135) | R@1 (90-135) | R@5 (90-135) | AUPRC (135-180) | R@1 (135-180) | R@5 (135-180) | AUPRC (Overall) | R@1 (Overall) | R@5 (Overall) |
> | - | - | - | - | -| - | -| - | - | - | - | - | - | - | - | - |
> | SAM2 (Video Prop) | **41.3** | **43.8** | **47.2** | 33.2 | 36.7 | 40.2 | 21.6 | 26.6 | 31.9 | 14.3 | 19.6 | 24.6 | 27.6 | 31.6 | 36.2 |
> | DINOv2-g14-AVG | 28.7| 35.0 | 45.2 | 20.9 | 30.2 | 43.6 | 16.2 | 26.8 | 42.3 | 15.4 | 25.8 | 41.7 | 20.3 | 29.4 | 43.2 |
> | MASt3R | 25.5 | 27.4 | 36.3 | 22.6 | 25.2 | 35.2 | 20.7 | 24.3 | 35.1 | 21.0 | 26.2 | 37.4 | 22.4 | 25.8 | 36.0 |
> | **SegMASt3R** | 40.4 | 42.7 | 46.4 | **39.0** | **41.8** | **45.8** | **37.7** | **41.2** | **45.8** | **37.6** | **41.5** | **46.3** | **38.7** | **41.8** | **46.1** |
>
> (continued in next comment to adjust for character limit)

---

> > ### Author Response · Authors · 2025-08-04
> >
> > ## DUSt3R Backbone
> > > Additionally, I would like to point out that DUSt3R is not trained for the matching task. The key distinction between DUSt3R and MASt3R is the incorporation of a matching loss.
> >
> > Correct! We would like to clarify the source of confusion: our statement “DUSt3R is trained for a matching task” was a reference to the use of ‘cross-attention’ (a pairwise internal comparison) in DUSt3R, but we agree with **jith** that a ‘matching loss’ is crucial for it to qualify as a ‘matching task’.
> >
> > > Therefore, using the DUSt3R backbone as a comparison to MASt3R's could help demonstrate the effectiveness of the explicit formulation of image matching.
> >
> > DUSt3R-SegFeat is exactly this! To avoid further confusion, we have now tabulated the modules (architecture/loss) used in the ablation study, as below:
> >
> > | Model             | Encoder     | Decoder (Cross-Attn) | Trained Head(s)                      | Loss Function(s)                     |
> > |------------------|-------------|-----------------------|------------------------------|--------------------------------------|
> > | **CroCo (Vanilla) [41, 42]**      | CroCo       | (unused during inference) | —                            | Cross-View Completion (SSL) |
> > | **CroCo-SegFeat (Ours)**        | CroCo       | —                     | Segment Feature Head                          | Segment Matching Loss                         |
> > | **DUSt3R (Vanilla)**     | CroCo       | ✅                     | PointMap Head                | Regression Loss                       |
> > | **DUSt3R-SegFeat**       | CroCo       | ✅                     | Segment Feature Head                            | Segment Matching Loss                         |
> > | **MASt3R (Vanilla) [21]**     | CroCo       | ✅                     | PointMap Head + Local Feature Head | Regression Loss + Matching Loss       |
> > | **SegMASt3R (Ours)**            | CroCo       | ✅                     | Segment Feature Head                            | Segment Matching Loss                         |

---

> > ### Comment · Reviewer_jith · 2025-08-05
> > **Thanks for the response**
> >
> > Thanks for the additional experiments and explanation. Most of my concerns have been addressed. One interesting observation is the large performance gap between using GT masks and FastSAM masks on ScanNet++—the numbers are halved. Do the authors have any comments on this? Does it indicate that the proposed method is sensitive to noisy masks?
> >
> > BR,
> >
> > jith

---

> ### Author Response · Authors · 2025-08-07
>
> Thanks for your response, **jith**.
>
> ## *Updated* Evaluation with Noisy Segments
>
> > the numbers are halved … Does it indicate that the proposed method is sensitive to noisy masks?
>
> No, **it is *not* the presence of noisy masks per se that affects performance, but their *misalignment* with GT masks during evaluation**. In other words, the lower numbers are due to an evaluation artefact than due to the matching performance of the proposed method. Our method consistently outperforms all baselines regardless of the quality of masks produced by the upstream Automatic Mask Generator (AMG - FastSAM here) used in the evaluation. Below, we explain our main evaluation strategy in more detail and also provide results through an additional mode of evaluation that decouples this *misalignment* from the models' matching performance.
>
> In our previous evaluation setup, the lower numbers compared to `Table 1` arise because they explicitly incorporate both “noisy segmentation” and “matching performance *conditioned* on those noisy segments”. AMG predicted (noisy) masks often do not perfectly align with GT masks (i.e., IoU < 1). To evaluate matching conditioned on these predicted segments, we employ an IoU threshold ($\theta$) that discards predicted matches corresponding to segments with IoU < $\theta$. In our primary evaluation, we used $\theta$ = 0.5, which reflects a moderate overlap. However, to perform a comparison inline with `Table 1` we evaluate against the full set of ground-truth correspondences, including those corresponding to predicted segments that were discarded due to not meeting the IoU threshold ($\theta$). This amounts to applying a penalty based on the *misalignment* between predicted and ground-truth segments in respective reference and query images.
>
> To isolate the effect of matching performance from this *misalignment* (i.e., the penalty in our previous evaluation setup), we now report results where low-IoU segments are excluded from evaluation *after* predicting matches - **this ensures that the matcher still sees noisy AMG inputs**.  Specifically, now during evaluation, we filter the ground-truth correspondences to only include entries for which at least one pair of predicted segments was retained after IoU thresholding. In the table below, titled “Evaluation *Without* Penalty”, it can be observed that our method maintains a clear lead in matching segments despite variations in segmentation noise. For comparison, we also report results for the “Evaluation *With* Penalty” scenario, which extends the results we shared in our previous response.
>
> ### Evaluation *Without* Penalty
> |Method| IoU > 0.25                  | IoU > 0.50                  | IoU > 0.75                  |
> |:--|:------------------------:|:------------------------:|:------------------------:|
> ||`AUPRC / R@1 / R@5`|`AUPRC / R@1 / R@5`|`AUPRC / R@1 / R@5`|
> | **SAM2** | 47.4 / 56.2 / 61.2| 50.2 / 58.7 / 62.6| 55.6 / 64.1 / 66.6|
> | **DINOv2**         | 39.0 / 60.0 / 89.2| 44.4 / 65.0 / 92.0| 54.9 / 73.1 / 94.0|
> | **MASt3R (Vanilla)**  | 50.7 / 58.0 / 76.2| 52.4 / 59.5 / 77.0| 49.6 / 57.4 / 76.3|
> | **SegMASt3R (Ours)** | **84.2 / 91.9 / 99.3**|**87.6 / 94.4 / 99.3**|**89.9 / 94.3 / 96.3**|
>
> ### Evaluation *With* Penalty
> |Method| IoU > 0.25                  | IoU > 0.50                  | IoU > 0.75                  |
> |:--|:------------------------:|:------------------------:|:------------------------:|
> ||`AUPRC / R@1 / R@5`|`AUPRC / R@1 / R@5`|`AUPRC / R@1 / R@5`|
> | **SAM2** | 31.5 / 36.7 / 41.6  | 27.6 / 31.6 / 36.2  | 18.3 / 20.7 / 25.0  |
> | **DINOv2** | 23.2 / 35.1 / 52.9  | 20.3 / 29.5 / 43.2  | 12.7 / 17.1 / 24.9  |
> | **MASt3R (Vanilla)**| 28.6 / 32.8 / 44.8  | 22.4 / 25.8 / 36.0  | 10.6 / 13.0 / 20.8  |
> | **SegMASt3R (Ours)**| **48.4 / 52.7 / 58.1**  | **38.7 / 41.8 / 46.1**  | **19.9 / 21.5 / 25.3**  |

---

> > ### Author Response · Authors · 2025-08-09
> >
> > We hope our reply on the noisy masks point was clear. Kindly let us know if there’s anything we can clarify before the discussion closes.

---

### Note · Authors · 2025-08-12

We thank the reviewers for acknowledging our core contribution in proposing a new baseline for wide-baseline segment matching (jith, mks5, vtn9, namx), in addition to proposing a method SegMaster that has effective model design (namx), with a pipeline that is simple, while very effective (vtn9). All reviewers acknowledged the thoroughness of the experimental evaluation and effectiveness of our method on the task of wide-baseline segment matching compared to baseline methods, as well as the downstream applications of instance mapping and robotic navigation  Additionally, the paper was itself acknowledged to be well structed and well written (mks5, vtn9, namx).

**Concerns addressed with reviewers’ satisfactory acknowledgment:** Generalization to outdoor data [jith, mKs5 vtN9], novelty [jith, mKs5, vtN9], requested ablation studies (DUSt3R backbone, decoder finetuning, learnable dustbin) [jith, nAMx], requested sensitivity analyses (feature resolution, training with non-GT masks, evaluating with non-GT masks), and other minor clarifications (learnable logit $\alpha$, abstract text, number of training segments M, and checkpoints).


**Concerns addressed BUT PENDING acknowledgement:** We respect reviewers’ time devoted to our paper but also regret their pending acknowledgements and further engagement to which we could have confidently provided clarification. Thus, we again clarify that our method a) [mks5] does not depend on 3D/instance annotation, b) [jith] is not sensitive to noisy (FastSAM) masks but was perceived so due to an evaluation artefact, and c) [mks5] is novel in terms of a new architecture (as a non-trivial composition), a newly established task (segment matching from narrow to wide baselines), and new knowledge, gained through empirical studies, i.e., feature behavior at pixel vs segment level, inverted backbone behaviors for off-the-shelf vs task-specific training, and relevance of geometric context and cross-attention.

---

### Decision · Program_Chairs · 2025-09-17

**Decision:**

Accept (spotlight)

**Comment:**

This paper introduces SegMASt3R, a novel method for wide-baseline segment matching that effectively leverages MASt3R to establish robust correspondences between semantic segments across extreme viewpoint changes. The paper's key strength is its compelling scientific insight: while a 2D model (DINOv2) outperforms a geometric one (MASt3R) off-the-shelf for this task, fine-tuning the geometric backbone with a simple segment-matching head yields state-of-the-art performance, proving that explicit geometric reasoning is essential. The authors' rebuttal was exemplary, thoroughly addressing all reviewer concerns with substantial new quantitative evidence on outdoor datasets and under noisy mask conditions, significantly elevating the paper's impact and demonstrating strong generalization and robustness. The additional ablations, including the critical role of the learnable dustbin, further strengthened the paper. All reviewers acknowledged that their concerns were fully satisfied, confirming the paper's technical soundness, clarity, and substantial contribution to the field. The combination of defining a new task, delivering a counter-intuitive finding. Therefore, the AC recommends acceptance.